# Different Doses of Carbohydrate Mouth Rinse Have No Effect on Exercise Performance in Resistance Trained Women

**DOI:** 10.3390/ijerph18073463

**Published:** 2021-03-26

**Authors:** Raci Karayigit, Scott C. Forbes, Alireza Naderi, Darren G. Candow, Ulas C. Yildirim, Firat Akca, Dicle Aras, Burak C. Yasli, Aysegul Sisman, Ahmet Mor, Mojtaba Kaviani

**Affiliations:** 1Faculty of Sport Sciences, Ankara University, Gölbaşı, Ankara 06830, Turkey; rkarayigit@ankara.edu.tr (R.K.); ulas.can.yldrm.ucy@gmail.com (U.C.Y.); fakca@ankara.edu.tr (F.A.); daras@ankara.edu.tr (D.A.); 2Department of Physical Education Studies, Brandon University, Brandon, MB R7A 6A9, Canada; forbess@brandonu.ca; 3Department of Sport Physiology, Boroujerd Branch, Islamic Azad University, Boroujerd 6915136111, Iran; Naderi_a@yahoo.com; 4Faculty of Kinesiology and Health Studies, University of Regina, Regina, SK S4S 0A2, Canada; darren.candow@uregina.ca; 5Faculty of Sport Sciences, Sinop University, Sinop 57000, Turkey; amor@sinop.edu.tr; 6Department of Physical Education and Sports, Igdir University, Iğdır 76000, Turkey; burakcaglar90@gmail.com; 7Faculty of Sport Science, Muğla Sıtkı Koçman University, Muğla 48000, Turkey; aysegulsisman@mu.edu.tr; 8School of Nutrition and Dietetics, Faculty of Pure & Applied Science, Acadia University, Wolfville, NS B4P 2R6, Canada

**Keywords:** ergogenic aid, female athletes, strength, muscular endurance

## Abstract

Carbohydrate (CHO) mouth rinse has been shown to enhance aerobic endurance performance. However, the effects of CHO mouth rinse on muscular strength and endurance are mixed and may be dependent on dosage of CHO. The primary purpose was to examine the effects of different dosages of CHO rinse on strength (bench press 1 repetition maximum [1-RM]) and muscular endurance (40% of 1-RM repetitions to failure) in female athletes. Sixteen resistance-trained females (age: 20 ± 1 years; height: 167 ± 3 cm; body mass: 67 ± 4 kg; BMI: 17 ± 2 kg/m^2^; resistance training experience: 2 ± 1 years) completed four conditions in random order. The four conditions consisted of a mouth rinse with 25 mL solutions containing either 6% of CHO (Low dose of CHO: LCHO), 12% CHO (Moderate dose of CHO: MCHO), 18% CHO (High dose of CHO: HCHO) or water (Placebo: PLA) for 10 s prior to a bench press strength and muscular endurance test. Maximal strength (1-RM), muscular endurance (reps and total volume), heart rate (HR), ratings of perceived exertion (RPE) and glucose (GLU) were recorded each condition. There were no significant differences in strength (*p* = 0.95) or muscular endurance (total repetitions: *p* = 0.06; total volume: *p* = 0.20) between conditions. Similarly, HR (*p* = 0.69), RPE (*p* = 0.09) and GLU (*p* = 0.92) did not differ between conditions. In conclusion, various doses of CHO mouth rinse (6%, 12% and 18%) have no effect on upper body muscular strength or muscular endurance in female athletes.

## 1. Introduction

Ingestion of carbohydrates (CHO) enhances endurance performance (>60 min). Mechanistically, CHO ingestion allows higher rates of CHO oxidation and maintenance of plasma glucose during exercise which spares muscle glycogen [1]. Despite sufficient glycogen stores to sustain high intensity exercise beyond 1 h, CHO ingestion has been shown to improve high intensity, short duration, cycling performance (<1 h), suggesting that CHO may alter performance via central mediated mechanisms [2]. Further, intravenous infusion of glucose, bypassing the oral cavity entirely, did not alter 1 h cycling time trial performance [3].

In 2004, Carter et al. [4] were the first to investigate CHO mouth rinsing using a 6.4% CHO solution rinsed for 5 s every 12.5% of a cycling test and found a significant improvement in 1 h cycling time trial performance (+2.8%) in elite endurance cyclists. While mechanisms were not measured, the authors speculated that afferent signals from taste-receptor cells housed in papillae in the tongue and spread out over the soft palate and larynx may have altered perceived exertion leading to greater exercise performance. Chambers, Bridge and Jones [5] utilized functional magnetic resonance imaging to demonstrate that both glucose (with sweetness) and maltodextrin (no sweetness) in the mouth activates brain regions, including the anterior cingulate cortex and striatum, which are important for reward and motor control. Since the first investigation by Carter et al. [4], several studies have demonstrated that carbohydrate mouth rinsing (CMR) can improve aerobic endurance [6,7] and high-intensity exercise performances [8,9]. However, most of the literature focuses on aerobic-endurance based exercise or running and sprint type activities [6,7,8,9,10,11,12,13,14]; yet, evidence on muscular strength and endurance is limited.

Reduced neural drive to skeletal muscles during resistance exercise is associated with fatigue [15] and to compensate, muscle electrical activity rises to maintain force output during exercise [16]. As such, purported central mediated effects of CMR may enhance resistance exercise performance. Presently, contrasting results have been reported with beneficial effects on bench press and back squat repetitions to failure at 60% of 1-RM [17], total training load volume [18,19], and isometric exercise [20] while several others have found no effects on maximal strength and endurance [21,22,23,24,25]. For example, a 6.4% CMR solution was used by Painelli et al. [25] and had no effect on bench press 1-RM or six sets to failure at 70% of 1-RM endurance performance. The authors postulated that decreasing the exercise intensity to a more muscular endurance oriented exercise may allow one to detect subtle benefits of CMR. The differences in outcomes across studies may be associated with methodological differences, including test duration, exercise selection, time of day, training and post prandial status of participants, doses of carbohydrate rinsed and sex. Further, the majority of these studies were conducted on males [17,19,20,21,22,24,25].

The caloric value but not sweetness of the carbohydrate solution is known to alter, likely in a dose response fashion, the stimulation of the reward center and motor control centers in the brain [5,26]. Since resistance exercise and endurance type activities have different central and metabolic responses, speculation can be made that carbohydrate receptors found in the remainder of the gastrointestinal tract (esophagus, upper intestine) needs to be stimulated by ingestion to find an improvement in high intensity efforts [27] such as resistance exercise but not for endurance exercise [28]. No data are available to date that investigates the effects of combined ingestion and mouth rinsing of carbohydrate on resistance exercise performance on females. Therefore, the aim of this study was to examine the effects of different doses of CMR on resistance exercise performance in female athletes. It was hypothesized that a CMR would augment resistance training performance in a dose response manner.

## 2. Materials and Methods

### 2.1. Participants

Sixteen healthy, non-smoking resistance-trained females volunteered to participate in the study (age: 20 ± 1 years; height: 167 ± 3 cm; body mass: 67 ± 4 kg; BMI: 17 ± 2 kg/m^2^; resistance training experience: 2 ± 1 years). Inclusion criteria included (1) having had no current or previous musculoskeletal injuries within the last year, (2) were performing resistance training (≥4 times per week) for the previous year and included bench press exercises in their training routine, and (3) were able to perform successful bench press exercise with a load corresponding to 100% of their current body mass. All participants declared that they were not using creatine, steroids or oral contraceptives, since these are known to alter muscle biology and muscular performance [29]. Written informed consent outlining the purpose, procedures and risks of the protocol was obtained from all participants, and the study procedures followed the principles outlined in the Declaration of Helsinki and were approved by Sinop University, Human Research Ethics Committee (decision no: 2021/29).

### 2.2. Study Design

Following the first session which was used as a familiarization, participants were randomized in a cross-over, counterbalanced, double-blind design, to four experimental conditions: mouth rinsing with a 6% carbohydrate solution (LCHO), 12% carbohydrate solution (MCHO), 18% carbohydrate solution (HCHO) or water as a placebo (PLA). Experimental sessions were conducted on different days (48 h apart). Bench press strength (1-RM) was assessed according to procedures described by Clarke et al. [21]. Muscular endurance performance was tested at 40% of their 1-RM until failure. Upon arrival at the laboratory, participants warmed up for 10 min on a treadmill followed by assessment of muscular strength and endurance, interspersed by a standardized 2 min rest period. Heart rate (HR) (Polar Team 2 telemetric system, Kemple, Finland), ratings of perceived exertion (RPE) (6–20 Borg) [30], and capillary glucose (GLU) (Accutrend Plus, Roche Diagnostics, Mannheim, Germany) were determined immediately after the endurance test, as shown in Figure 1. Blood sample were taken from a fingertip. Participants completed all sessions in the morning (08:00–10:00) following a 10 h overnight fast. The rinsing solutions were coded by a non-affiliated researcher to ensure double blinding. Participants were asked to avoid caffeine and alcohol intake and vigorous exercise in the 24 h leading up to each visit. Further, a 24-h dietary intake was recorded, and participants were asked to replicate it prior to each experimental condition. Diet was also confirmed verbally prior to each testing session.

### 2.3. Strength (1-RM) and 40% of 1-RM Muscular Endurance Test Protocol

To warm up, participants performed 10 repetitions with a light weight (20 kg) followed by resting for 1 min and a further 3–5 repetitions with adding 10% more weight followed by 2–3 repetitions approaching near maximum effort with 2 min rest. Participants were then provided a 3 min rest interval prior to their first 1-RM attempt. If successful, the weight was increased ~10% with a minimum of 3 min rest, otherwise weight was decreased by 2.5–5%. As previously described [21,24], upper-body strength performance was measured with bench press in 3–5 attempts. Once 1-RM was achieved a 2 min passive rest was given to adjust the weight for 40% of 1-RM, which was used for the muscular endurance test. Participants completed as many repetitions as possible to failure using this load. To standardize bench press technique, a certified personal trainer checked the protocol and provided feedback throughout. During both strength and muscular endurance tests, a metronome was followed to control repetition speed at 2 s for eccentric and concentric phases. The researchers were only allowed to intervene and stop the protocol if there was a failure to perform additional repetition with proper technique and posture.

### 2.4. Mouth Rinsing Protocol

Immediately prior to every 1-RM test, and before every muscular endurance test, participants rinsed their mouth with their respective experimental solution. The solutions were provided in a plastic cup, and the participants were instructed to swish the beverage in the buccal cavity for 10 s and then spat it into the cup. Solutions were water as a placebo (PLA), 6% low dose (LCHO), 12% moderate dose (MCHO) and 18% high dose (HCHO) of maltodextrin and were taste-matched and colourless and composed of 300 mg of sucralose indifferent in appearance. The same researcher mixed the solutions using electronic laboratory scales the day before each experimental trial using water at room temperature.

### 2.5. Statistical Analysis

All data were analyzed using the IBM SPSS statistics for Windows, version 22.0 (IBM Corp., Armonk, NY, USA). Strength, muscular endurance performance and RPE were analyzed using a one-way repeated analysis of variance (ANOVA) and HR, GLU were analyzed using a two-way repeated measures (condition x time) ANOVA. If significant interactions or main effects were observed, pairwise comparisons with a bonferroni correction was applied. The data were reported for each dependent variable as mean ± SD, with an alpha level of *p* < 0.05. Sphericity was analyzed by Mauchly’s test of sphericity followed by the Greenhouse-Geisser adjustment where required. To assess test–retest consistency of the four conditions, intraclass correlation coefficients (ICC) were computed. The effect sizes were calculated using partial eta squared (η^2^) as trivial (<0.10), moderate (0.25–0.39), or large (≥0.40).

## 3. Results

No significant differences in 1-RM were observed between conditions (F_3,45_ = 0.106, *p* = 0.95, η^2^ = 0.01) and showed high ICC (0.94) (Figure 2).

Furthermore, muscular endurance for total repetitions (F_3,45_ = 2,533, *p* = 0.06, η^2^ = 0.14; Figure 3A) and total volume (F_3,45_ = 1.588, *p* = 0.20, η^2^ = 0.09; Figure 3B) were not significantly different between conditions. Repetition numbers and total volume (repetitions × load) showed high ICC (0.96 and 0.94, respectively).

Heart rate significantly increased after exercise (F_1,15_ = 1250.400, *p* = 0.01, η^2^ = 0.98), but there were no significant differences between conditions (F_3,45_ = 0.478, *p* = 0.69, η^2^ = 0.03). Glucose responses were not significantly different between conditions (F_3,45_ = 2.281, *p* = 0.09, η^2^ = 0.13). Similarly, RPE did not differ between conditions (F_3,45_ = 0.154, *p* = 0.92, η^2^ = 0.01), as shown in Table 1.

All data presented mean and standard deviation; Pre-test: prior to the resistance exercise protocol; Post-test: Immediately after the exercise protocol.

## 4. Discussion

The main finding of the present study was that rinsing with a low (6%), moderate (12%) or high (18%) dose of carbohydrate has no meaningful effect on upper body resistance exercise performance (strength or endurance). Furthermore, the composition of the solution has no effect on glucose, heart rate or ratings of perceived exertion.

The dose–response relationship of CMR on resistance exercise performance has not been investigated before, thus direct comparisons in this regard are not possible, however there have been a few dose response studies examining aerobic endurance performance. The findings of the current study support those of Kulaksız et al. [31] who showed that 3%, 6% and 12% maltodextrin mouth rinsing failed to improve 20 km cycling time trial performance, RPE or HR in recreationally active males. The authors recognized that the VO_2_max of the participants were lower (21–42%) than the average CMR literature [31]. Ispoglou et al. [32] also demonstrated that 4%, 6% and 8% of CMR had no improvement in aerobic endurance in the postprandial state (had a meal 3 h before exercise) with either concentration of carbohydrate solutions compared to placebo. Although it was shown by James et al. [5] that mouth rinsing with a 7% maltodextrin solution for 5 s routinely during exercise improved ~1 h cycling time trial performance, no dose response (7% vs. 14%) effect was reported.

Presence of carbohydrate in the oral cavity may attenuate centrally mediated inhibition of motor output possibly through afferent signals associated with stimulation of brain areas responsible for reward and motor control [5]. In support of this assumption, increases in motor evoked potentials and cortico-motor output to the exercised muscles involved in elbow flexion in both non-fatigued and fatigued states with CMR was demonstrated by Gant et al. [33]. However, CMR did not affect upper body muscular performance in the current study even with a high dose (18%). Previous studies support the current findings that 6% to 18% CHO solutions did not increase bench press muscular endurance (repetitions to failure) performance [21,22,24]. In contrast, Clarke et al. [17] reported a 6% CHO solution significantly increased repetitions to failure at 60% of 1-RM after a series of exercise tests such as countermovement jumps, isometric mid-thigh pull and a 10 m sprint. Apparently, maximal muscle strength and strength endurance performance (measured by 1-RM test) are not influenced by CMR [21,23,25] particularly in a non-fatigued state; therefore, using more sensitive methods such as isokinetic dynamometer should be employed to detect small improvements (1–3%) with CMR in future investigations.

Some of the previous studies showed benefits of CMR compared to water-placebo in the absence of fatigue [8,9], such as a single set of upper-body muscle endurance at 40% of 1-RM intensity, which was the same intensity applied in the present study, however, our results demonstrated no improvement. Despite a limited ecological validity, Jensen et al. [20] reported that CMR resulted in decreased torque attenuation as compared with a noncaloric control mouth rinse, in a fatigue state. However, Green et al. [23] investigated CMR on 60% of 1-RM bench press repetitions to failure after 4 sets of 10 repetitions at 65% of 1-RM with a 2-min recovery between sets in a moderate fatigue state and found no increase in bench press performance with CMR compared to placebo in both male and female resistance trained participants. Despite no observed sex-based difference in that study, the authors suggested that preferential effect of a CMR can be expected to a greater extent in males [23]. Krings et al. [22] also reported that 6% of CMR did not improve repetitions to failure and total session volume of upper-body resistance exercises at 70% of 1-RM (bench press, bent-over row, incline bench press, close-grip row, hammer curls, skull crushers, push-ups and pull-ups) compared to placebo (carbohydrate: 203 vs. placebo: 201 reps). The authors suggested, based on the use of only upper body exercises, using larger muscle groups (lower body) would potentially see benefits from CMR. In support, neuromuscular performance and corticomotor excitability of the rectus femoris was enhanced with a 6.4% of CMR [34]. Furthermore, Durkin et al. [35] showed that squat but not bench press repetition to failure performance at 40% of 1-RM was increased with 6% of CMR with low carbohydrate availability. Further research is needed to investigate the high dose response of CMR between upper and lower body in the fatigued and non-fatigued or low and high muscle glycogen state.

Despite no statistical increase in repetitions during 3 sets of each half-squat, bench press, military press and seated row exercises, 6% of CMR resulted in 12% greater total training volume compared to placebo in recreationally trained women [18]. It can be suggested that CMR may enhance muscle performance depending on the duration of the training session, since improvement was shown following whole-body exercise lasting approximately 50 min [17,18] compared to 20 min [23,25] and about 10 min as in the present study. However, others have reported that 6% of CMR was found to be ineffective on resistance exercise performance with session duration approximately 50 min [22]. Exercise selection, test protocol, supplementation doses and participants training status might explain the various responses across the current and previous investigations. More research is therefore warranted.

The current study sought to address the limitations of the previous studies presented in the literature [21,22,23,24,25] by stabilizing tempo of the movement (2 s for both concentric and eccentric phases). In the present study, 6%, 12% and 18% of CMR was compared to a non-carbohydrate placebo (water) condition and no improvements were found in bench press strength and endurance performance. Since the areas of the brain such as cingulate cortex and ventral striatum were unresponsive to artificial sweetener [5], purported benefits of CMR can be expected to be apparent over water-placebo. However, even small amounts of water in the mouth significantly improved exercise time in a dehydrated state [36] and heart rate [37] possibly through activation of pharyngeal receptors. In this regard, 6.4% of maltodextrin mouth rinsing for 10 s was shown to significantly increased bench press repetition numbers compared to no-rinse control condition (+1.4) but not to a water-placebo condition (+1.1) [19]. Further, 10 km time trial performance post-30 min steady-state cycling (at 65% VO_2peak_) and post-test RPE values improved with mouth rinsing, regardless of the composition of the mouth rinse solution (either carbohydrate or water-placebo) compared to no-rinse control condition [38]. In the current and previous studies [22,23,39] no increase in resistance exercise performance with mouth rinsing, absence of a no-rinse control condition can be a reason to not finding improvements. Although CMR was reported by a few studies [21,24,25]. Future research should include a no-rinse control condition to detect subtle effects of either water and carbohydrate or both of them synergistically with a dose–response manner.

The current investigation is not without limitation. Although the fixed 48-h was given between conditions for each participant (whole experiment lasted ~1 week), menstrual cycle was not measured and may affect the synaptic potentiation associated with corticomotor excitability [40]. Due to the low estrogen concentration prevented enhanced corticomotor excitability by 6.4% of CMR in females [13], further research needs to examine whether magnitude of CMR’s effect on resistance exercise performance vary in different menstrual cycle phases. Although participants were asked to repeat their 24-h diet prior to each test day, daily calorie intake was not analyzed, and glycogen stores may differ between sessions that may affect resistance exercise performance.

## 5. Conclusions

Mouth rinsing with 6%, 12% or 18% CHO solutions did not enhance upper body strength or muscular endurance performance in trained females. Further, heart rate, glucose and ratings of perceived exertion did not differ between CHO conditions and placebo. Sinop University, Human Research Ethics Committee (decision no: 2021/29).

## Figures and Tables

**Figure 1 ijerph-18-03463-f001:**
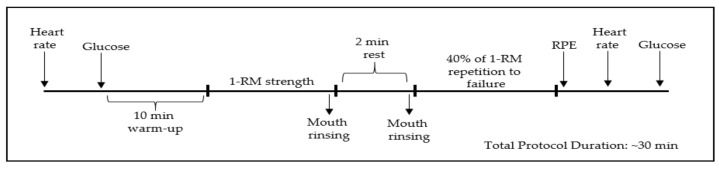
Schematic diagram of experimental procedures.

**Figure 2 ijerph-18-03463-f002:**
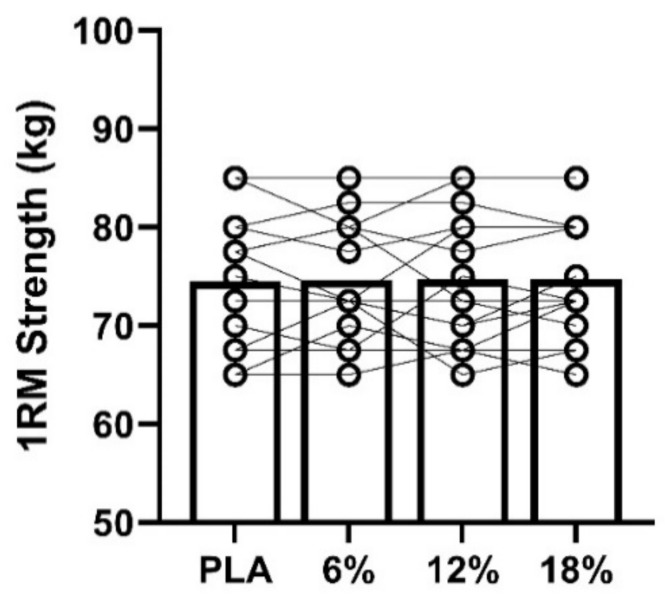
1-RM strength (kg) across the four conditions.

**Figure 3 ijerph-18-03463-f003:**
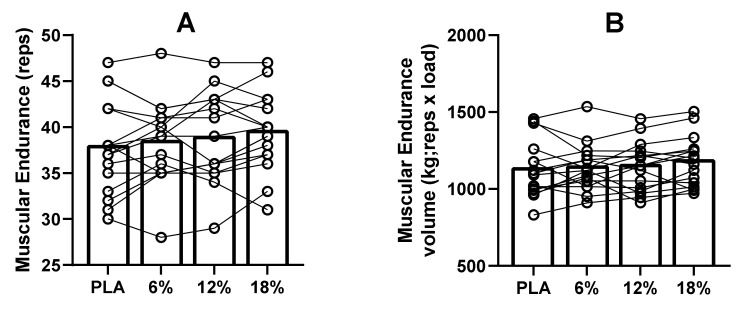
Muscular Endurance total repetitions (**A**) and total volume (**B**).

**Table 1 ijerph-18-03463-t001:** Heart rate, glucose and ratings of perceived exertion before and after the exercise protocol in each condition.

	PLA		LCHO		MCHO		HCHO	
	M	SD	M	SD	M	SD	M	SD
	**Heart Rate (Beat/min)**							
Pre-Test	65.5	5.2	67.3	4.7	67.1	3.5	66.3	4.3
Post-Test	153.0	12.8	149.3	10.9	152.0	11.4	152.6	14.4
	**Glucose (mg/dL)**							
Pre-Test	85.6	6.5	88.5	6.4	91.0	5.8	90.8	5.9
Post-Test	90.4	9.7	91.5	9.1	93.6	9.6	93.6	9.3
	**Rating of Perceived Exertion (RPE) (6–20)**							
Post-Test	16.6	2.0	16.9	2.2	16.8	2.6	16.9	2.4

PLA: Placebo, LCHO: Low dose of Carbohydrate, MCHO: Moderate dose of Carbohydrate, HCHO: High dose of Carbohydrate, SD: Standard Deviation, M: Mean.

## Data Availability

The data presented in this study are available on request from the corresponding author. The data are not publicly available due to restrictions privacy.

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
