# Peer review of "Different Doses of Carbohydrate Mouth Rinse Have No Effect on Exercise Performance in Resistance Trained Women"

_ijerph, 2021, doi:10.3390/ijerph18073463_

Round 1

Reviewer 1 Report

The authors have presented an interesting paper. Overall, the manuscript is well prepared. After carefully reading the text, the following comments occurred to me that the authors should consider and possibly use to strengthen the paper.

  1. The introduction seems inconsistent and needs strengthening. The authors emphasize that there has been no similar study to date and at the same time cite a study by Painelli et al. A quick review of the database indicates several publications on similar topics:

Dunkin, J. E., & Phillips, S. M. (2017). The effect of a carbohydrate mouth rinse on upper-body muscular strength and endurance. The Journal of Strength & Conditioning Research, 31(7), 1948-1953.

Bastos-Silva, V. J., Prestes, J., & Geraldes, A. A. (2019). Effect of carbohydrate mouth rinse on training load volume in resistance exercises. The Journal of Strength & Conditioning Research, 33(6), 1653-1657.

Krings, B. M., Shepherd, B. D., Waldman, H. S., McAllister, M. J., & Smith, J. W. (2020). Effects of carbohydrate mouth rinsing on upper body resistance exercise performance. International journal of sport nutrition and exercise metabolism, 30(1), 42-47.

Clarke, N. D., Kornilios, E., & Richardson, D. L. (2015). Carbohydrate and caffeine mouth rinses do not affect maximal strength and muscular endurance performance. The Journal of Strength & Conditioning Research, 29(10), 2926-2931.

In addition, the authors claim that "Additionally, the ergogenicity of mouth rinsing can vary between sexes during resistance exercise training" yet the study involved only women, the authors cannot verify this hypothesis in their paper.

In the introduction, the authors focus on the effect of CMR on endurance performance during long-lasting exercise > 1h. These efforts are probably based on aerobic metabolism using glucose, whereas in this paper they examine short efforts based probably mainly on anaerobic metabolism (e.g. phosphagen). The authors should justify in the introduction the hypothesis that CMR may also affect this kind of efforts.

2) Figure1: please add the total duration of the exercise protocol. This is important.

3) How can the study be blind if placebo is water vs. liquid with carbohydrate (sweet?)

4) How will the authors interpret the post-exercise glucose increase despite the exercise? The results lack information on whether these changes were significant. This should be discussed.

5. In the discussion, comparison of the results to similar studies but with prolonged exercise is unjustified. These efforts have a different metabolic background than the efforts analyzed in the authors' study. The effect of CMR may be different. Please focus on the effect of CMR on efforts with similar metabolism.

Reviewer 2 Report

Thank you for providing a well written paper and well thought out experiment to review. The design is simple, and the findings are well presented. I would like some more detail on their interpretation and xxx. Please find below a line by line review:

Introduction

58 - 60 - you use however twice in subsequent sentences here. Please revise. The second however, may be replaced by whereas or yet, or even with.

71/72 - fed status of participants is also likely a determinant of efficacy here, that has implications for the present work given your participants were fasted. This is discussed most recently by Rollo et al., (2020) and may be considered to affect secondary or tertiary outcomes of performance.

73 - I'd consider using 'the majority' here, as opposed to 'the vast majority' given you're only briefly detailing 7 studies

76 - Please revise this statement. The Chambers study did not directly assess the effects of calorie content from carbohydrate containing beverages upon exercise performance or mechanisms. The studies within this work assessed the effect of sweetness (glucose vs maltodextrin) and taste-matched carbohydrate content (glucose vs saccharine). These experiments are quite different to the effects of energy density, but may confer signals as to energy availability (Best et al., 2020; Berthoud, 2003; Breslin, 2013; Reed & Knaapila, 2010).

78 - the subsequent mechanistic interpretation is interesting, but I don't know if it's strong enough to warrant suggesting that sex differences in response to mouth rinsing may occur. This may simply be an issue of ordering. If you put the mechanism first, demonstrate that this mechanism may be influenced by mouth-rinsing, and then link back to the mechanism as a rationale for conducting this work in females, that should work. At present the statement is too strong, to stand alone.

93/94 - please provide a reference or two to support this statement. The effects in literature on oral contraceptive use for e.g. upon strength performance are mixed.

101 - please replace the full stop following familiarization with a comma.

It may be worth clarifying that all solutions were colourless, as this has been shown to affect placebo treatments and ergogenic strategies in medicine, sports science and sports nutrition research - Szabo has written on this, I believe and is also covered in the placebo section of the review on taste by Best, cited above.

110 - which scale was used to assess RPE? Please reference

113/114 - just a reminder, it is worth mentioning fed/fasted in the introduction, as you manipulate it here.

128 - 'as many repetitions as possible to failure using this load'

129/130 - what is meant by feedback here? May this not affect performance in some way? I agree maximal effort and attainment of 'true failure' is important here, but it may be worthy of consideration here and subsequent discussion if the authors wish to clarify

153 - please could you add interpretations/ descriptors for eta squared here? This is important, given your results focus solely on p-values as the arbiter of significance, despite results that for some individuals show a great deal of variation. Allowing readers to interpret the effect sizes you've performed would allow for richer results and discussion sections and may elucidate some of this variability.

It may also be an idea to report a typical variability of the 1RM test and/or fatigue test, as a %CV or similar, that way we can make a more complete interpretation of the results, and assess whether any changes between solutions exceed this threshold and so are beyond that which can be typically expected to occur. This avoids the use of p-values as the sole arbiter of 'statistical truth'. I must add, excellent to see the ICC used in this manner.

As per the above points muscular endurance presented with a p-value of 0.06 and Figure 3A clearly shows a high degree of variability for some participants. A moderate/medium effect is seen here, when eta squared is interpreted. Similarly, the same holds for glucose between concentrations. If interpreted as a moderate effect, this supports your introductory point regarding energy availability/density and stands to reason in a fasted population.

 175/178 - based on the above outlined factors, I do not feel that these conclusions are supported. They do not convey the nuance of your findings accurately. This is a good study, with some interesting findings, and some nuance - celebrate your results!

200-204 - you may find some of the work on bitter tastants interesting too, as mechanistically these align to power activities to a greater extent than CHO. This has been referenced by Pickering (2019), cited in Best et al., (2020) and performed largely by Gam et al., in a series of works.

212 - bench press

Is there a possibility of saturation of oral carbohydrate receptors at higher concentrations? i appreciate 6% is not a higher concentration, but it may be a plausible explanation for 18% - perhaps there is a 'sweet spot' for mouth swill concentration, pardon the pun.

242/243 - please amend to fatigued and non-fatigued

263 - please clarify in a dehydrated state for reference [35]. You may also find that the paper at the following link provides some useful context that may support your explanation here - https://www.mdpi.com/2306-5710/7/1/9

272/274 i strongly agree with this comment, nice job. Evidence for the importance of this can be found within the following study - https://www.mdpi.com/2075-4663/8/6/90

278/280 - please rephrase this sentence, at present it sounds a little overly academic and the meaning is lost because of this. Keep it direct.

In summary, a well conducted experiment with a couple of revisions to undertake. I'd love to see more nuanced description of your results in the discussion, focussing on the magnitude of observed effects beyond statistical significance alone, would enhance the paper greatly and better contextualise the present findings - this has been done to some extent with the mention of percentage performance enhancement (line 245).

References:

Berthoud H-R (2003) Neural systems controlling food intake and energy balance in the modern world. Current Opinion Clinic Nutr Metab Care 6:615–620

 Best, R.; McDonald, K.; Hurst, P.; Pickering, C. Can taste be ergogenic? European Journal of Nutrition 2020, 1–10, doi:10.1007/s00394-020-02274-5.

Breslin, P.A.S. An Evolutionary Perspective on Food Review and Human Taste. Current Biology 2013, 23, R409–R418, doi:10.1016/j.cub.2013.04.010.

Reed, D.R.; Knaapila, A. Genetics of Taste and Smell. In; Genes and Obesity; Elsevier, 2010; Vol. 94, pp. 213–240 ISBN 9780123750037.

Round 2

Reviewer 1 Report

Dear Authors, thank you for your responses. I have some suggestions, I believe will improve your paper.

  1. L78-82 I understand that you wanted to emphasise that women will be studied, but in my opinion this is not necessary in this paper - no men were studied and this section of the introduction suggests that both men and women will be studied as there may be different effects. Please remove.
  2. “We believe that the introduction sets the stage for why we believe it may enhance performance. The first studies were on endurance exercise, but the mechanism appears to be centrally mediated and thus relevant to resistance exercise, hence the flow of our introduction”

Agreed, the mechanism may be central, but the introduction still lacks data to support your hypothesis that CMR may relate to resistance exercise. Just because it affects endurance does not mean it will affect resistance exercise. There is a large disparity between the description of the effect of CMR on endurance and on resistance exercise. Please introduce in the introduction a possible mechanism of CMR influence on resistance exercise. Currently it looks like this: CMR has an effect on endurance, next the previous data on this topic are presented. There are only two sentences supporting your hypothesis in the whole introduction. Please elaborate.

  1. In discussion section do not discuss the effects of CMR on endurance – this is not related to your study (L200-L220), focus on resistance performance. Delete this paragraph – maybe it is good for intro but not for discussion section.

Author Response

We would like to thank the reviewer for their valuable and helpful comments/suggestion on our manuscript. We have now addressed all the comments in the revised manuscript. A point by point responses are outlined below.

  1. Thanks for this suggestion. Per your suggestion, line 78-82 has been removed.
  2. We have now added another sentence to elaborate more on the possible  mechanisms by which CMR may impact resistance performance. 
  3. Per your suggestion, we have now removed this paragraph and move some lines again per your suggestion up to the introduction.